# Highly anisotropic $Fe_3C$ microflakes constructed by solid-state phase transformation for efficient microwave absorption

Rongzhi Zhao [1,2,3], Tong Gao[1,2,3], Yixing Li [1,2] ✉, Zhuo Sun[2], Zhengyu Zhang[2], Lianze Ji[1], Chenglong Hu[1], Xiaolian Liu[1], Zhenhua Zhang[1], Xuefeng Zhang [1,2] ✉ & Gaowu Qin [2]

Soft magnetic materials with flake geometry can provide shape anisotropy for breaking the Snoek limit, which is promising for achieving high-frequency ferromagnetic resonances and microwave absorption properties. Here, two-dimensional (2D) $Fe_3C$ microflakes with crystal orientation are obtained by solid-state phase transformation assisted by electrochemical dealloying. The shape anisotropy can be further regulated by manipulating the thickness of 2D $Fe_3C$ microflakes under different isothermally quenching temperatures. Thus, the resonant frequency is adjusted effectively from 9.47 and 11.56 GHz under isothermal quenching from 700 °C to 550 °C. The imaginary part of the complex permeability can reach 0.9 at 11.56 GHz, and the minimum reflection loss ($RL_{min}$) is −52.09 dB (15.85 GHz, 2.90 mm) with an effective absorption bandwidth ($EAB_{\leq -10 \ dB}$) of 2.55 GHz. This study provides insight into the preparation of high-frequency magnetic loss materials for obtaining high-performance microwave absorbers and achieves the preparation of functional materials from traditional structural materials.

Electromagnetic (EM) interference has emerged as a crucial problem with the rapid development of high-frequency communication techniques and micro-miniaturization of electronic information devices, which has aroused the rapid development of EM wave absorption materials[1–6]. Among all these achievements, soft magnetic materials occupy an important position in designing materials because of their excellent magnetic loss abilities induced by high saturation magnetization ($M_s$)[5,7,8]. However, their ferromagnetic resonance behaviors could be impeded by the contradictory relationship between loss ability and frequency, known as the Snoek limit ($(\mu_i − 1)f_r = 4\gamma M_s/3$)[9]. Consequently, dielectric materials such as carbon, MXene, and ceramic have been introduced to obtain satisfactory EM wave absorption at gigahertz[6,10,11]. However, an approach to cross the Snoek limit by manipulating the ferromagnetic materials themselves is still lacking, thus limiting the development of high-frequency magnetic loss materials and the optimization of composited materials.

Shape anisotropy, which is constructed through controllable preparation processes, has been recognized as a promising pathway to address this shortcoming[12]. The flake geometry could induce an effective in-plane anisotropy in soft magnetic materials to break the Snoek limit, which could optimize the magnetic loss ability at gigahertz[13–17]. Under these guidelines, various soft magnetic materials, such as flake-like $FeNi_3$[18], $ZnCo_2O_4$ crystalline nanosheets[19], flake-like $Sm_{1.5}Y_{0.5}Fe_{17−x}Si_x$[20], $Ce_2Fe_{17}N$ microflakes[13], and carbonyl iron microflakes[21], have been investigated, in which their ferromagnetic resonance frequencies have been extended to gigahertz regions. Nevertheless, it should be noted that their loss abilities, reflected by the imaginary part of the complex permeability ($\mu''$), still exhibit a decreasing tendency as the frequency increases to the

[1]Institute of Advanced Magnetic Materials, College of Materials and Environmental Engineering, Hangzhou Dianzi University, Hangzhou 310012, China. [2]Key Laboratory for Anisotropy and Texture of Materials (MOE), School of Materials Science and Engineering, Northeastern University, Shenyang 110819, China. [3]These authors contributed equally: Rongzhi Zhao, Tong Gao. ✉e-mail: liyx@mail.neu.edu.cn; zhang@hdu.edu.cn

high-frequency region. For example, the $\mu''$ values of $Sm_{1.5}Y_{0.5}Fe_{17}Si$, $FeSiAl/MnO_2$, and $FeSiAl/ferrite$ are 0.6, 1.2, and 1.05 at 5.0, 5.0, and 6.0 GHz, respectively. The values of $Sm_{1.5}Y_{0.5}Fe_{15.5}Si_{1.5}$, barium ferrite, and $FeSiAl/Al_2O_3$ ceramics were reduced to 0.38, 0.31 and 0.48 at 9.5, 9.1 and 9.4 GHz[20,22–25]. Therefore, a soft magnetic material with high ferromagnetic resonance frequency and magnetic loss capacity is urgently needed to meet the requirements of high-frequency applications.

Cementite ($Fe_3C$) with high saturation magnetization can be an ideal candidate to achieve the above requirements because of its high saturation. However, there remains a great lack of controllable synthesis of morphology through traditional chemistry methods because the structures of $Fe_3C$ are difficult to restrict by coating agents. Here, we propose a pathway via solid-state phase transformation to obtain the large-scale preparation of $Fe_3C$ microflakes with crystal orientation. The morphology can be adjusted by solid-state phase transformation, and the material can be obtained by electrochemical

dealloying at a constant voltage (CV) of −0.4 V. The electromagnetic response performances and theoretical simulations indicate that the ferromagnetic resonance frequency relies on the increased effective in-plane anisotropy induced by the flake geometry, which can be regulated by adjusting the isothermal quenching temperatures of eutectoid steels. As a result, the 2D $Fe_3C$ microflakes treated at 550 °C could achieve the highest anisotropy, which can achieve an excellent magnetic loss ability ($\mu''$) of 0.9 at a frequency of 11.56 GHz. Thus, our results clarify the relationship between anisotropy and high-frequency magnetic loss ability, proposing a new route for designing high-performance microwave absorption materials.

## Results

According to the Fe−C binary phase diagram and the isothermal quenching curves of eutectoid steel (0.77 wt%[C]) (Fig. 1a), the microstructures of pearlite can be changed by the isothermal quenching process within the temperature regions of 550 °C to the temperature of

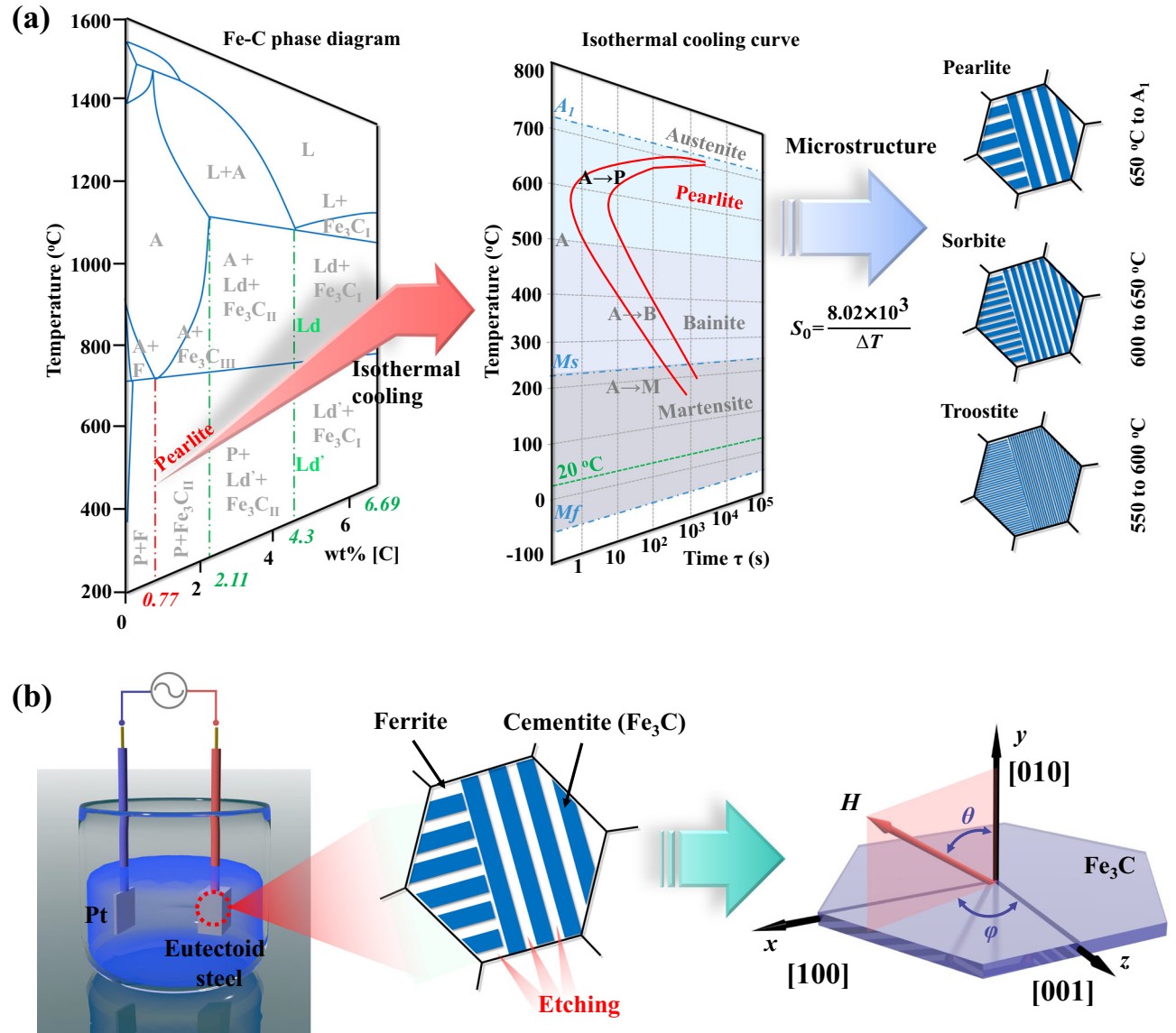

**Fig. 1 | Synthesis process (design route of solid-state phase transformation) of $Fe_3C$ microflakes. a** Fe−C binary phase diagram and isothermal quenching curve of eutectoid steel and schematic illustrations of pearlite, sorbite, and troostite, and the white and blue stripes represent ferrite and $Fe_3C$, respectively. (A: austenite, B: bainite, F: ferrite, L: liquid, Ld/Ld': ledeburite, M: martensite, P: pearlite, $M_f$: martensite finish, $M_S$: martensite start, $S_0$: average interlamellar distance) (**b**) Schematic illustrations of the electrochemical dealloying process and the morphology of $Fe_3C$ microflakes, and the $Fe_3C$ microflake (right) is derived from the blue cementite band (middle) after the dealloying process.

eutectoid transformation (marked by $A_1$)[26–28]. The average interlamellar distance $S_0$ of different pearlite microstructures, so-called pearlite (650 °C to $A_1$), sorbite (600–650 °C), and troostite (550–600 °C), can be evaluated by the empirical formula of $S_0$ (nm) = $8.02 \times 10^3/\Delta T$ ($\Delta T$ refers to the difference between the heat treatment temperature (800 °C) and the isothermal quenching temperature)[27,28], which indicates that the thicknesses of cementite can be adjusted by designing the isothermal temperatures. In this context, the original eutectoid steels were first cut into a cuboid with a size of $20 \times 10 \times 5$ mm$^3$ and annealed at 800 °C for 3 h to completely austenitize. Then, the eutectoid steel precursors were placed into the molten salt to realize the isothermal quenching process, during which the austenite transformed into pearlite, sorbite, or troostite. Finally, the precursors were set as the electrode and connected to the electrochemical workstation in CV mode (−0.4 V). The 2D Fe$_3$C microflakes, denoted as Fe$_3$C−$x$ ($x$ stands for quenching temperature), were prepared by electrochemical dealloying of the ferrite textures within a solution of KCl and C$_6$H$_5$Na$_3$O$_7$[29], as illustrated in Fig. 1b. The corresponding current curves of the electrochemical dealloying process with decreasing tendencies can be recognized, as shown in Supplementary Fig. S1.

As shown in Fig. 2a, a typical pearlite structure can be observed in all the samples using scanning electron microscopy (SEM), in which the bright and dark stripes are the cementite and ferrite textures, respectively. The corresponding thicknesses of cementite were collected and summarized in Fig. 2e and Supplementary Fig. 2. The thicknesses decreased from 22.0 to 15.7 nm as the quenching temperatures decreased from 700 to 500 °C, implying that the Fe$_3$C structures can be manipulated by the isothermal quenching process. The microstructures of the eutectoid steel precursors after the dealloying process (Fe$_3$C-700) were characterized by SEM and shown in Supplementary Fig. 3. It can be seen that the ferrite has been etched, and the precursor was constructed by the flake structural cementite. Subsequently, the microstructures of the 2D Fe$_3$C microflakes were characterized by SEM and transmission electron microscopy (TEM), as shown in Fig. 2b, c. Here, an irregular flake structure could be observed in all the samples. It can be noticed that the thickness of Fe3C is nanoscale, but the width and length are microscale, which results in a high aspect ratio for reinforcing the shape anisotropy to enhance natural resonance performance. The elemental distributions and crystalline structures of Fe$_3$C were detected by energy-dispersive X-ray

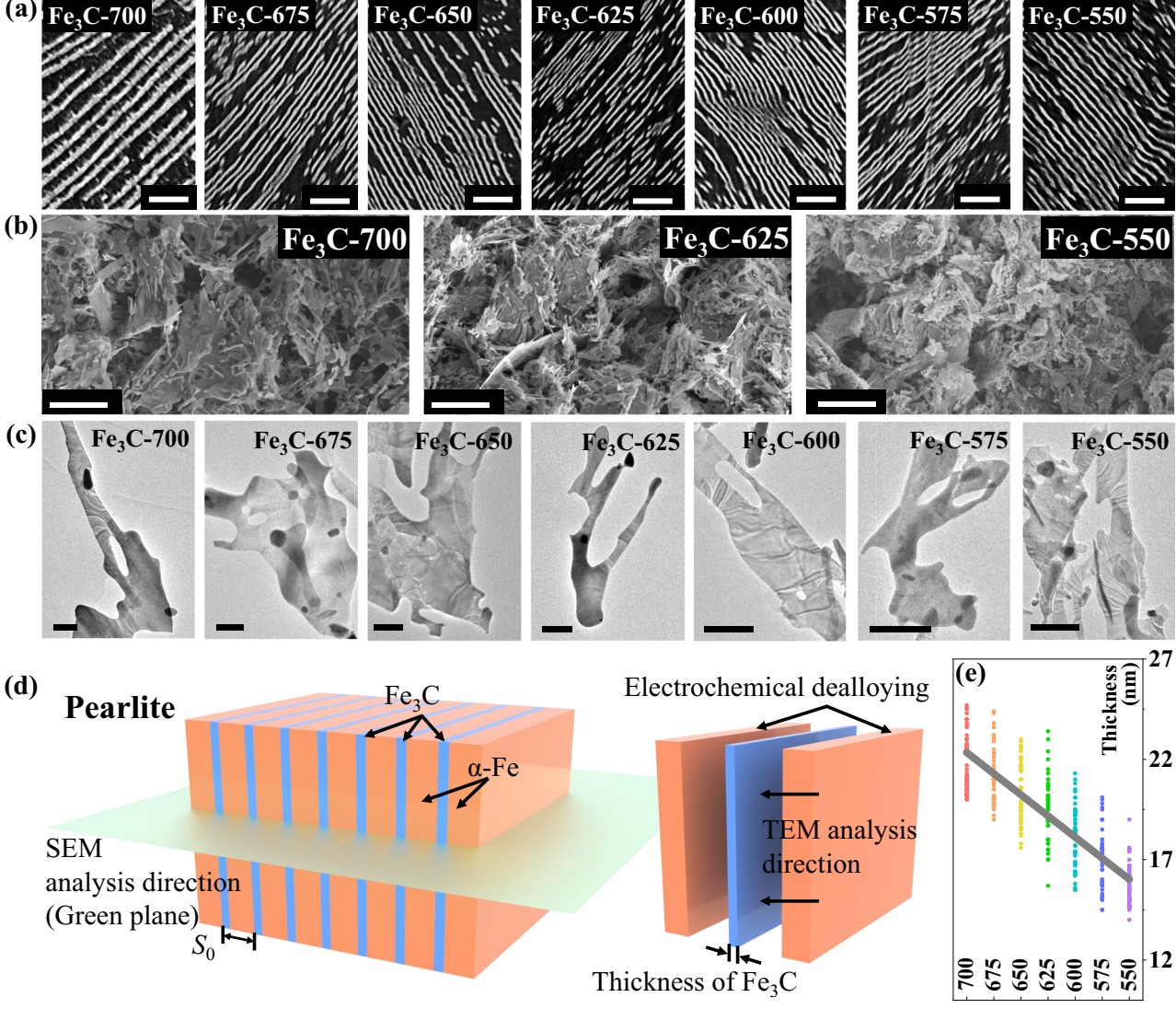

**Fig. 2 | Microstructure characterizations. a** SEM images of eutectoid steel isothermal quenching under different temperatures, scale bar: 2 µm. **b** High-resolution SEM images of the Fe$_3$C-700, Fe$_3$C-625, and Fe$_3$C-550 micro-flakes, scale bar: 10 µm. **c** High-resolution TEM image, atomic EDS maps, and SAED image of Fe$_3$C-700 microflakes, scale bar: 1 µm. **d** Schematic diagram of pearlite, electrochemical dealloying process, and analysis directions. **e** Summarization of the thickness of Fe$_3$C microflakes (Fitting line: $y = 2.024 - 0.00003x$, $R^2 = 0.698$). The number refers to the isothermal quenching temperature, and source data are provided as a Source Data file.

spectroscopy (EDS) and X-ray diffraction (XRD), respectively (Supplementary Figs. S4 and S5). The schematic diagram of pearlite has been exhibited in Fig. 2d and the corresponding definition of thickness has been displayed. Two composed elements, carbon, and iron, were uniformly distributed in the $Fe_3C$ microflakes, and their phase structures were highly identical to the standard phase card PDF#85-1317[30,31], indicating that the 2D $Fe_3C$ microflakes were successfully obtained through the electrochemical dealloying process.

Using spherical aberration-corrected TEM, we further investigated the microstructures of these 2D $Fe_3C$ microflakes, in which high-resolution TEM (HR-TEM) images of $Fe_3C$-700, $Fe_3C$-625, and $Fe_3C$-550 were obtained and shown in Supplementary Fig. 6. As the atomic EDS mapping demonstrated, the iron and carbon atoms are uniformly posited in the 2D microflakes. The enlarged view of HR-TEM images and the corresponding inverse fast Fourier transform (IFFT) images (Supplementary Fig. S6B) further disclose the lattice structures of $Fe_3C$-700, in which the red spots represent iron atoms and the blue spots are carbon. It can be found that the zone axis of all the $Fe_3C$ is [010], which has been identified by the selected area electron diffraction (SAED) images shown in Supplementary Figs. S6A and S7[32,33]. It is concluded that the crystal structure of 2D $Fe_3C$ microflakes is orthorhombic with the space group of *Pnma*. The magnetocrystalline anisotropy can be inferred, where the easy axis of [001] is located in the plane of 2D $Fe_3C$ microflakes and the second easy axis of [010] is located out-of-plane[34]. Thus, the intrinsic in-plane anisotropy introduced by the solid-state phase transformation assisted by electrochemical dealloying can effectively improve the magnetic loss ability in the high-frequency region.

The magnetic performances of all the samples were analyzed through the hysteresis loops, as exhibited in Supplementary Figs. S8 and S9. The saturation magnetizations ($M_s$) are 97.7, 103.5, 114.6, 119.9, 109.2, 128.3, and 125.8 emu/g, respectively, and the coercivities ($H_c$) are 279.6, 271.8, 194.4, 114.4, 106.5, 104.2, and 69.6 Oe. The $M_s$ increased with the enhanced thicknesses of $Fe_3C$ microflakes, while the $H_c$ exhibited an inverse phenomenon. The $Fe_3C$-550 microflakes presented the highest $H_c$ among all the samples. Because the coercivity is proportional to the anisotropy constant[35], it can be inferred that the increase in coercivity is the consequence of the increased anisotropies of the 2D $Fe_3C$ microflakes. Thus, the shape anisotropy can be efficiently manipulated by adjusting the isothermal quenching conditions[36].

The microwave response performances of all the $Fe_3C$ microflakes were measured by a vector network analyzer (VNA) in the frequency region of 2–18 GHz, as demonstrated in Supplementary Figs. S10 and S11. The measurements of each sample were repeated more than 5 times. In general, the dielectric loss ability should not be observed in the single magnetic loss material. In all the $Fe_3C$ microflakes, it can be seen that the complex permittivity values of the real parts ($\varepsilon'$) are approximately centered at 20 and the imaginary parts ($\varepsilon''$) are centered at 0. However, some resonance peaks can be noticed in the complex permittivity patterns, both in real and imaginary parts. To clearly demonstrate the position of these resonance peaks, the superimposed figure of complex permittivity and complex permeability has been employed and shown in Supplementary Fig. 12. It can be noticed that the values can be found in two areas: the high-frequency turning point of the real part, and both ends of the resonance peak of the imaginary part. Such a phenomenon is mainly associated with the transformation between the permittivity and permeability, which is realized by the connection between the $Fe_3C$ microflakes because of their high anisotropy microstructures[5]. According to studies such as close-packed Ni nanoparticles, multi-walled carbon nanotubes, and sub-nanometer clusters in nanocages, the aggregation of conductive medium would construct the conductive network, resulting in a reinforced polarization behavior for increasing the dielectric loss ability[5,37,38]. Therefore, the irregular

shape of $Fe_3C$ microflakes would form a conductive network in the EM testing sample, which can induce polarization around the natural resonance and thus result in the vibration of complex permittivity.

From the results of the complex permeability, typical ferromagnetic resonances can be found in all the samples, and their frequencies are within the regions of 9.47–11.56 GHz. The complex permeability of all the samples was fitted through the Landau–Lifshitz–Gilbert (LLG) equation based on five different experimental results, and the details of the fitting process have been demonstrated in the method section. Here, an increased resonance peak ($f_r$) can be observed along with the decreased isothermal temperature (Fig. 3a, b). It can be noticed that the $f_r$ of $Fe_3C$-700 microflakes, containing the lowest magnetic anisotropy of all the samples, can even reach 9.73 GHz, which is far higher than that of previously reported soft magnetic materials such as c-axis oriented *hcp*-(CoIr) thin films (4.5 GHz)[39], $Fe_{20}Ni_{80}$ and $Co_{20}Fe_{60}B_{20}$ material-modulated stripe-patterned thin films (5.8 GHz)[40], and $Sm_{1.5}Y_{0.5}Fe_{17-x}Si_x$ and their composites (5.0 GHz)[20]. The results indicate that the magnetic loss abilities can be significantly optimized in the 2D $Fe_3C$ microflakes by adjusting the isothermal temperature, ascribed to the manipulation of shape anisotropies. In the optimized $Fe_3C$-550 microflakes, which present the highest anisotropy, $f_r$ can be enhanced to 11.56 GHz (Fig. 3c). For the magnetic loss ability (imaginary part of the complex permeability, $\mu''$)[6], the values of all the $Fe_3C$ microflakes reached 1.05, 1.27, 1.43, 0.86, 0.82, 1.00, and 0.90 at 9.58, 10.07, 10.24, 10.33, 10.87, 11.00, and 11.56 GHz, respectively. The magnetic loss abilities of $Fe_3C$ microflakes can be effectively improved in high-frequency regions compared with the $Sm_{1.5}Y_{0.5}Fe_{15.5}Si_{1.5}$ (0.38, 9.1 GHz)[20], ferrites (0.31, 9.1 GHz)[22], and FeSiAl (0.48, 9.4 GHz)[24]. Furthermore, to visually illustrate the optimized magnetic loss abilities, the pattern of $f_r$ versus $\mu''$ of $Fe_3C$ microflakes was compared to the other reported magnetic loss materials in Fig. 3d (Supplementary Refs. 1–26), in which the $Fe_3C$ samples with increased anisotropy could possess satisfactory magnetic loss performances at gigahertz among the related reports. Furthermore, the magnetic loss angles (tan $\delta_M$)[41] were calculated by the complex permeability to further imply the loss performances of $Fe_3C$ microflakes (Fig. 3e). Similar to the complex permeability, the loss abilities were improved along with the increased anisotropy.

The microwave absorption properties of the 2D $Fe_3C$ microflakes were evaluated by transmit-line theory[42–44], as shown in Fig. 3c and Supplementary Fig. 13. The minimum reflection loss ($RL_{min}$) values of $Fe_3C$ are −52.09, −44.39, −42.02, −52.82, −45.73, −43.29, and −56.99 dB, respectively, in which −10 and −20 dB indicate that 90% and 99% incident EM energy can be attenuated[42,43,45]. For the effective absorption bandwidth ($EAB_{\leq-10\ dB}$), the values of 2.55, 2.60, 2.75, 3.05, 2.60, 3.75, and 2.69 GHz with thicknesses of 1.20, 2.60, 1.25, 1.25, 1.25, 1.10, and 1.35 mm can be observed. The 2D $Fe_3C$ microflakes exhibit excellent microwave absorption performance only through their magnetic loss abilities. For the impedance-matching performances of $Fe_3C$ microflakes, the values of $|Z_{in}/Z_0|$[6] were collected and are shown in Supplementary Fig. 14. Commonly, the values in the regions of 0.8 and 1.2 should be considered as excellent impedance-matching performances[6,46]. Here, the 2D $Fe_3C$ microflakes could obtain satisfactory matching performances in the entire frequency region (2–18 GHz), indicating that the manipulation of shape anisotropy could simultaneously improve the impedance matching and microwave absorption performances in high-frequency regions.

Therefore, it is concluded that a high-frequency magnetic loss ability can be achieved in $Fe_3C$ microflakes with enhanced shape anisotropy. To further identify the details of such performances, Lorentz transmission electron microscopy (L-TEM) was employed, and the corresponding in-focus (Fig. 4a, b) and overfocus (Fig. 4c, d) images are shown, in which strip domain structures can be observed in the $Fe_3C$-700 microflakes. The nucleation of strip domains can result from the appearance of intrinsic out-of-plane anisotropy, which is similar to the manipulation of magnetic domains in sputtered (or evaporated)

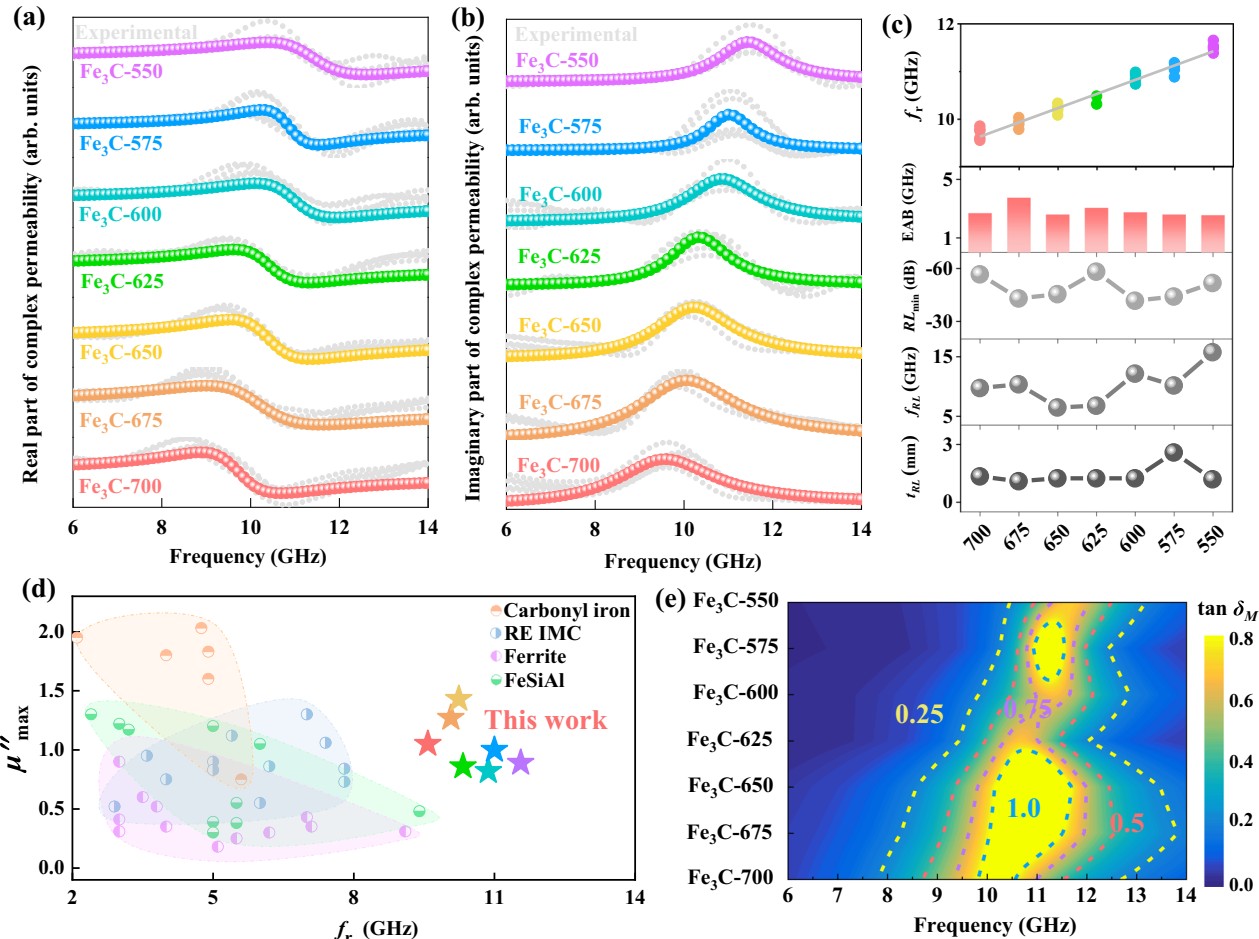

**Fig. 3 | Magnetic loss abilities. a** Real part ($\mu'$) and **b** imaginary part ($\mu''$) of the complex permeability at the frequency regions of 6–14 GHz, in which the colored fitted lines are fitted from the experimental lines (gray lines) through the Landau–Lifshitz–Gilbert (LLG) equation. **c** Summarization of the natural resonance frequencies (Fitting line: $y = 9.348 + 0.299x$, $R^2 = 0.956$), effective absorption bandwidth, reflection loss and the corresponding thickness of Fe$_3$C. **d** Comparison of $f_r$ versus $\mu''$ of reported soft magnetic materials and Fe$_3$C. **e** The tan $\delta_M$ of Fe$_3$C microflakes in the frequency region of 6–14 GHz. Source data are provided as a Source Data file.

magnetic thin films by adjusting the perpendicular anisotropy[47,48]. The strip domain structures under a static magnetic field ($H_{ext} = 0$ mT) can be reconstructed by micromagnetic simulations with mumax3[49] by considering the shape anisotropy as an effective in-plane anisotropy ($K_i$), as shown in Fig. 4e. The color denotes the x-component of magnetization. The simulated results are consistent with the calculated distribution of magnetization from contrast L-TEM images by the classic transport of intensity equation (TIE) method[50] (see Fig. 4f). Thus, the ferromagnetic resonance behaviors of Fe$_3$C microflakes under different temperatures can also be obtained from the static magnetic domains by solving the Landau–Lifhiz–Gilbert equation numerically.

Figure 5a shows the ferromagnetic resonance spectra of the Fe$_3$C microflakes under different temperatures after obtaining the static magnetic structures with different values of $K_i$, which are induced by the shape anisotropy of Fe$_3$C microflakes with different thicknesses. The resonant frequencies of the simulated Fe$_3$C microflakes increase from 15.1 GHz to 17.8 GHz with the increase in $K_i$ from $5 \times 10^4$ J/m$^3$ to $7 \times 10^4$ J/m$^3$ with an interval of $0.5 \times 10^4$ J/m$^3$, in which the chosen values of $K_i$ are considered for fitting the change in coercivity (Supplementary Figs. S8 and S9) and the change in frequency is consistent with the experimental results. The difference between the simulation and experimental values of frequency could result from the defect in a real material system, where the ideal material model is considered in micromagnetic simulations. Thus, it is inferred that the increased

resonant frequencies can be ascribed to the increase in effective in-plane anisotropy induced by flake geometries, which can be manipulated by adjusting the isothermal quenching temperatures. Furthermore, the amplitude distribution of resonance at 15.1 GHz for the Fe$_3$C-700 microflakes is shown in Fig. 5b, c, which is focused on the position of domain walls and named localized spin wave modes. Therefore, the absorption of EM for Fe$_3$C microflakes at high frequency could originate from the appearance of large numbers of domain walls in stripe domains. The corresponding phase distribution of the Fe$_3$C-700 microflakes is also shown in Fig. 5d, e. The phase is almost uniform in a single domain wall, which is similar to the propagation of spin waves vertically along the domain wall.

## Discussion

In summary, 2D anisotropic Fe$_3$C microflakes with crystal orientations were prepared from eutectoid steel precursors through solid-state phase transformation assisted by electrochemical dealloying. It can be recognized that the high-frequency ferromagnetic resonance behavior is strictly correlated to the shape anisotropy, which can be manipulated by adjusting the isothermal quenching temperatures. The stripe domain structures are observed directly without a biased magnetic field using in situ L-TEM, which can be reconstructed by micromagnetic simulations with in-plane and out-of-plane anisotropies. Consequently, the natural resonance frequencies of Fe$_3$C microflakes have been increased to the regions of 9.47–11.56 GHz, while the mostly

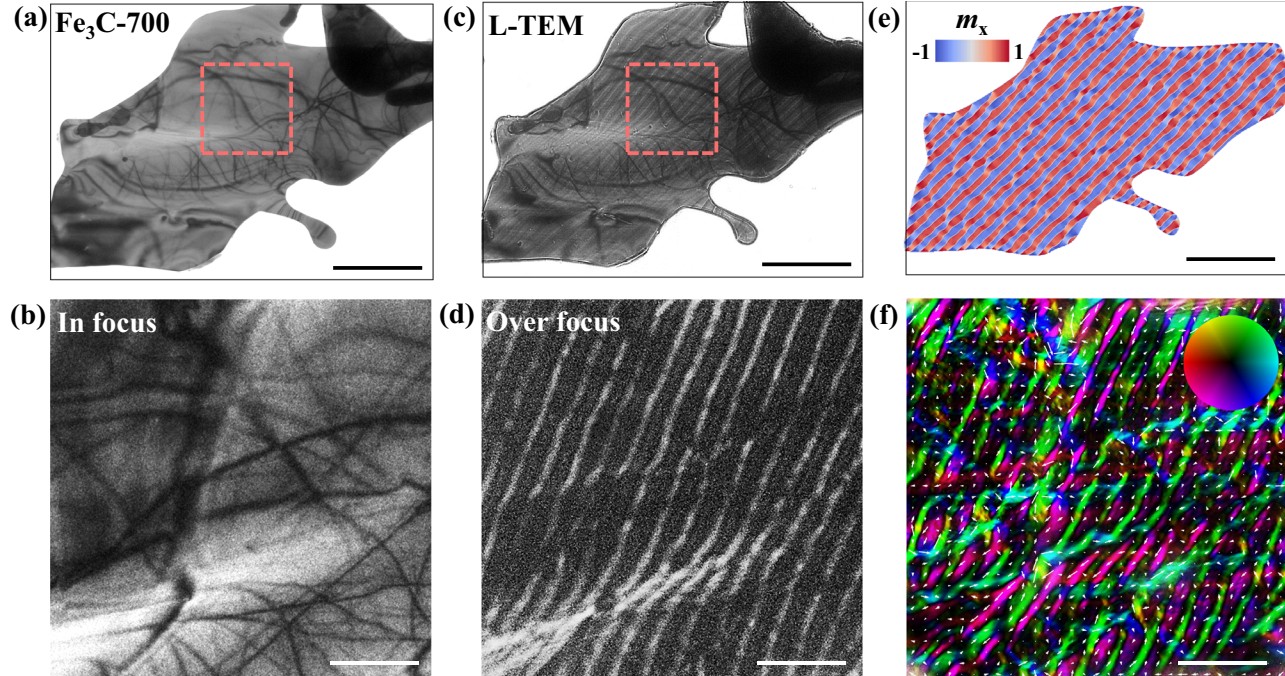

**Fig. 4 | L-TEM characterizations of Fe₃C-700 microflakes. a** In-focus L-TEM images and **b** enlarged view of the red dashed square, scale bar: 500 nm. **c** Overfocus L-TEM images (scale bar: 2 μm) and **d** enlarged view of the red dashed square (scale bar: 500 nm). **e** Simulated distribution of magnetization by micromagnetic simulations, scale bar: 2 μm. **f** Calculated distribution of magnetization from contrast L-TEM images by the classic transport of intensity equation (TIE) method, and the color denotes the x-component of magnetization, scale bar: 500 nm. Source data are provided as a Source Data file.

optimized Fe₃C-550 microflakes can reach 11.56 GHz with a $\mu''$ value of 0.9. Moreover, the RL_min and EAB_≤−10 dB of Fe₃C-550 microflakes could reach −52.09 dB (15.85 GHz, 2.90 mm) and 2.55 GHz (1.20 mm), respectively, exhibiting an optimized microwave absorption property. The present study provides an intrinsic insight into the role of anisotropy in high-frequency magnetic loss ability and provides a new route to design high-performance microwave absorption materials. Furthermore, this method can stimulate research interests in obtaining functional materials from other traditional structural materials. We believe that this work also provides some preliminary research basis for the realization of structural–functional integrated materials.

## Methods

### Electrochemical dealloying for synthesizing Fe₃C micro-flakes

The eutectoid steel (0.77 wt%[C]) was firstly cut into the blocks with the size of $20 \times 10 \times 5$ mm³ and annealed at 800 °C for 3 h in the ambient atmosphere. Then, the blocks were placed into the molten salt for 30 min to suffer the isothermal quenching process. Here, the isothermal temperature was achieved by two different salt groups (wt%): (i) 30% KCl + 20% NaCl + 50% BaCl₂ for 700 °C and 675 °C, (ii) 30% NaCl + 32% BaCl₂ + 48% CaCl₂ for 550 °C to 650 °C. After that, the blocks were fixed as an electrode in the solution of KCl and C₆H₅Na₃O₇ to engage the electrochemical dealloying process for 48 h. The electrochemical dealloying process was obtained by an electrochemical workstation working in constant voltage (CV) mode with a voltage of −0.4 V. The as-made Fe₃C micro-flakes were washed with the mixture of HCl (4%) and ethanol (96%) and the deionized water for several times and then dried in the vacuum drying oven for 12 h.

### Structural characterizations

The scanning electron microscope (SEM) was characterized by the JEOL JSM-IT500HR/LA with an accelerating voltage of 20 kV. The microstructure and morphology of the micro-flakes were characterized by the field-emission transmission electron microscope (FE-TEM) (Thermo Scientific Talos F200S), and the energy-dispersive X-ray spectroscopy (EDS) was obtained by Bruker Super Lite X2. The high-resolution TEM images were obtained by the spherical aberration-corrected transmission electron microscope (JEOL ARM-200). The statistic of the micro-flake thickness is calculated via the SEM images using the open-access *ImageJ* software. The crystalline structure of Fe₃C micro-flakes was calculated by the open-access *VESTA* software. The magnetic properties were measured via the vibrating sample magnetometer (VSM, ADE EV9) with a maximum applied field of 15 kOe. The Lorentz transmission electron microscope (L-TEM) images under a static magnetic field were analyzed by Thermo Scientific Talos F200S. X-ray diffraction (XRD) was performed using a SmartLab9kW (Rigaku) with a scan step of 0.04°.

### Electromagnetic response performances

The electromagnetic response performance was obtained by the vector network analyzer (VNA, N5222A, Keysight Co., Ltd.) equipped with a Type-N 50 Ω coaxial airline (2–18 GHz, Ceyear CETC). The VNA was calibrated by the 2-port short-open-load-thru (SOLT) calibration standard (85054B), and the samples were uniformly mixed with the paraffin at a weight percentage of 50 wt% and then processed to a toroidal shape of 7.00 mm outer diameter and 3.04 mm inner diameter. The reflection loss (RL) and efficient absorption bandwidth (EAB) were calculated via the transmit-line theory using the following equations (S3 and S4)[6,43].

$$R(\text{dB}) = 20\lg\left|\frac{Z_{\text{in}} - 1}{Z_{\text{in}} + 1}\right| \tag{1}$$

$$Z_{\text{in}} = \sqrt{\frac{\mu_r}{\varepsilon_r}}\tanh i\frac{2\pi f}{c}\sqrt{\mu_r \varepsilon_r}d \tag{2}$$

In with $Z_{\text{in}}$ is normalized input impedance, $c$ is the light velocity in the free space, $f$ is the frequency, and $d$ is the thickness of the absorber.

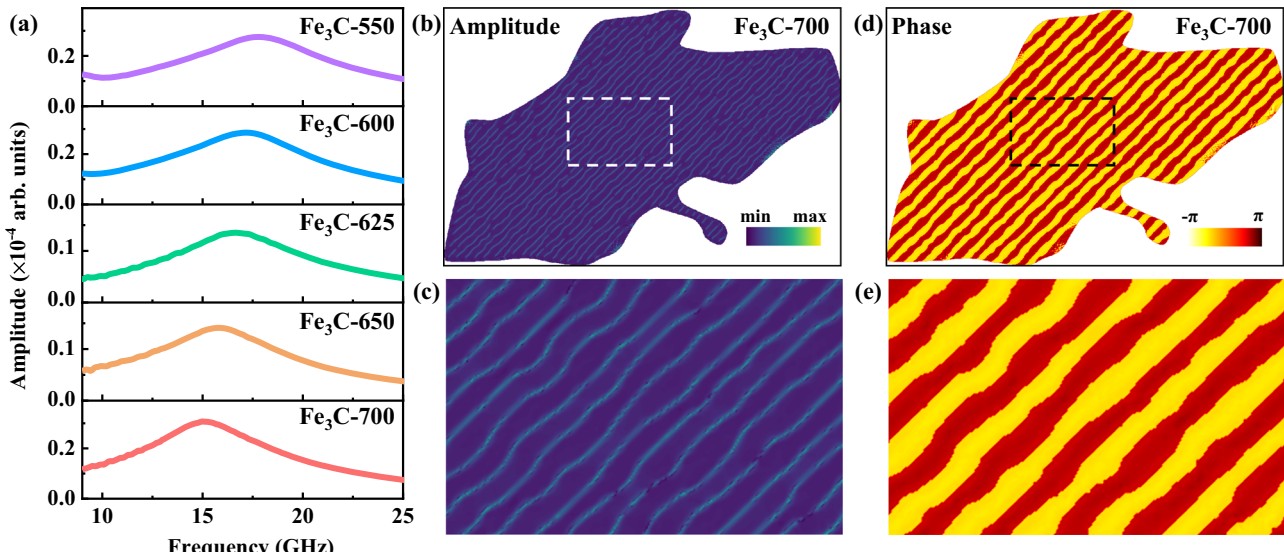

**Fig. 5 | Simulated ferromagnetic resonance behaviors of Fe₃C microflakes.**
**a** Simulated results of the imaginary part of the complex permeability for Fe₃C microflakes under different temperatures. **b** Amplitude distribution of resonance at 15.1 GHz for the Fe₃C-700 microflakes and **c** enlarged view of the region in the white dashed square. **d** Phase distribution of resonance at 15.1 GHz for the Fe₃C-700 microflakes and **e** enlarged view of the region in the black dashed square. Source data are provided as a Source Data file.

## Fitting process of complex permeability

The fitting of the complex permeability of all the samples has been processed by OriginPro 2021 (9.8.0.200, OriginLab Corporation). All the fitting line is fitted by five different experimental lines, and the frequency region was set at 6–14 GHz. The complex permeability can be fitted based on the classical LLG as follows[51,52]:

$$\mu' = 1 + \chi_0 \frac{1 + (\alpha^2 - 1)\left(\frac{f}{f_r}\right)^2}{\left[1 - (1+\alpha^2)\left(\frac{f}{f_r}\right)^2\right]^2 + \left(2\alpha\frac{f}{f_r}\right)^2} \quad (3)$$

$$\mu'' = \chi_0 \frac{\alpha\left(\frac{f}{f_r}\right)\left[1 + (1+\alpha^2)\left(\frac{f}{f_r}\right)^2\right]}{\left[1 - (1+\alpha^2)\left(\frac{f}{f_r}\right)^2\right]^2 + \left(2\alpha\frac{f}{f_r}\right)^2} \quad (4)$$

Where $\chi_0 = \mu_i - 1$ is the initial susceptibility, $\alpha$ is the damping constant and $f$ is the operating frequency. During the fitting process, the $\chi_0$, $\alpha$, and $f_r$ have been set as $a$, $b$, and $c$, and the variates including $\mu''$, $\mu'$, and $f$ have been employed to fit the constant $a$, $b$, and $c$. In addition, the natural resonance frequency $f_r'$ can be obtained through the equation as follows[13]:

$$f_r' = \frac{f_r}{\sqrt{1+\alpha^2}} \quad (5)$$

Where the $f_r$ and $\alpha$ are obtained from the above fitting process.

## Theoretical calculations

The micromagnetic simulations were conducted using the mumax3[49]. The material parameters are the following: The saturation magnetization is $M_s = 8 \times 10^5$ A/m. The exchange constant $A = 6 \times 10^{-12}$ J/m. The anisotropy constant of out-of-plane ($K_z$) and in-plane ($K_i$) are $K_z = 1 \times 10^5$ J/m³ and $K_i = 5 \times 10^4$ J/m³, respectively. The thickness is set as 20 nm. The damping constant is $\alpha = 0.1$, which is obtained by averaging the fitted damping constant in Fig. 3b, as shown in Supplementary Table 1. It should be noted that the $\alpha$ in the

simulation usually affects the intensity of amplitude in the position of resonant peaks, which does not affect the position of resonant peaks (see Supplementary Fig. 15)[53]. Thus, in micromagnetic simulations, the influence of anisotropy on ferromagnetic resonances is studied by fixing $\alpha$ at 0.1. The mesh size is $10 \times 10 \times 2$ nm³. The static magnetic structures were obtained by minimizing the total energy ($E_t$) of the magnetic system, including Heisenberg exchange energy ($E_{ex}$), anisotropy energy ($E_{ani}$), demagnetization energy ($E_{deg}$), and Zeeman energy ($E_{zeem}$)[54]. The magnetization dynamical behaviors in the magnetic system were obtained by solving the Landau–Lifshitz–Gilbert (LLG) equation:

$$\frac{d\boldsymbol{M}}{dt} = -|\bar{\gamma}|\boldsymbol{M} \times \boldsymbol{H}_{eff} - \frac{|\bar{\gamma}|\alpha}{M_s}\boldsymbol{M} \times (\boldsymbol{M} \times \boldsymbol{H}_{eff}) \quad (6)$$

where $\bar{\gamma}$ is the gyromagnetic constant, $\boldsymbol{M} = (M_x, M_y, M_z)$ denotes the magnetization vector, $\boldsymbol{H}_{eff}$ is the effective field and can be calculated by $H_{eff} = \frac{\partial E_t}{\partial \boldsymbol{M}}$. A perpendicular sinc field $h_z$ was applied to excite the dynamics of the magnetic system and was given by the following:

$$h_z(t) = h_a \sin[2\pi f_c(t - t_0)]/[2\pi f_c(t - t_0)] \quad (7)$$

where $t_0 = 10$ ps and $f_c = 50$ GHz. The field amplitude $h_a$ is fixed at 5 mT. The fast Fourier transform (FFT) of $M_z/M_s$ was used to obtain the ferromagnetic resonance spectra. The amplitude and phase distributions were obtained by performing the FFT of every mesh in the magnetic system, where the intensity at 15.1 GHz for the Fe₃C-700 microflakes was plotted in the $x$–$y$ plane.

## Data availability

The data that support the findings of this study are provided in the main text and the Supplementary Information. The original data are available from the corresponding author upon request. Source data are provided with this paper.

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

## Acknowledgements

The authors gratefully acknowledge the National Science Fund for Distinguished Young Scholars [No. 52225312 (X.F.Z.)], the National Natural Science Foundation of China [Nos. 52201202 (Y.X.L.) and U22A20117 (Y.X.L.)] and the Natural Science Foundation of Zhejiang Province of China [Nos. 2019C01121 (X.F.Z.) 2021C01033 (X.F.Z.)].

## Author contributions

Y.X.L., X.F.Z. and G.W.Q. conceived the idea and designed the present work. R.Z.Z., T.G. and Z.S. carried out the preparation. Z.Y.Z., R.Z.Z, L.Z.J., C.L.H., X.L.L. and Z.H.Z. carried out the TEM characteristics. Z.Y.Z., T.G. and Y.X.L. performed the microwave absorption measurements. R.Z.Z. and Y.X.L. performed the theoretical simulation. All authors discussed the results and contributed to the final paper.

## Competing interests

The authors declare no competing interests.
