## [Peer Review File · Nature Communications]

REVIEWERS' COMMENTS

Reviewer #1 (Remarks to the Author):

This study presents the design of Fe₃C microflakes with high anisotropy through solid-state phase transformation, with potential application in microwave absorption. Generally speaking, this work is interesting and has some novelties in the materials design and the property simulations. However, there are many unclear and/or ambiguous expressions in the content, likely due to a rushed submission without sufficient preparation and polishing. The simulations are critical in understanding the effects of thickness on Fe₃C microflakes, but unfortunately, the relevant details on the simulation process are largely absent. It is therefore unable to evaluate the validation of the simulations. Negative ϵ'' values can be observed in Figure S13, and most of the ϵ' values of the samples vary a lot. However, the author claims that the values of the real part (ϵ') are approximately 20 and the imaginary part (ϵ'') is 0. It seems like the declaration is not true and may mislead readers. In addition, there are many obvious mistakes or typos in the manuscript. The manuscript is not well written and structured, and the corresponding mechanism is insufficiently supported by the questionable points in simulation process and experimental results. A rejection is suggested considering the high standard of this journal.

Some specific comments:

1. The resonance peaks in Figure 3a and b shifts a lot in the diluted (experimental) line for some of the sample, for example, Fe₃C-600 and Fe₃C-700. Please, provide a reasonable explanation. In addition, the details on the diluted plotting should be included in the figure caption,
2. Figure 4f lacks of scale bar. What is the "colored ball" locating in top right corner of figure 4f? Please make it clear.
3. The authors should provide the detailed information (including the model, equations etc.) of the simulation in figure 5B-C, which may help to understand and evaluate the simulations.
4. The authors claim that the simulation in figure 5B-C could prove the absorption of EM for Fe₃C microflakes at high frequency could originate from the appearance of large numbers of domain walls in stripe domains. However, in my opinion, the authors should further clarify the influence of difference anisotropy by comparing the amplitudes of the samples fabricated by different temperatures, which may support the results in Figure 5a.
5. The damping constant of 0.1 was used for the simulations in figure 5a. Why chose 0.1? Was it from the electromagnetic parameters of Fe₃C-700?
6. Figure caption of figure 5 seems not right, and the scale bars in the figures are missing.
7. Negative ϵ'' values can be observed in Figure S13, and most of the ϵ' values of the samples vary a lot. As a result, the following declaration is questionable: 'For the complex permittivity (Figure S13, Supporting Information), it can be seen that the values of the real part (ϵ') are approximately 20 and the imaginary part (ϵ'') is 0. (Page 9 in the manuscript)

Reviewer #2 (Remarks to the Author):

In this paper, the Fe₃C with different shape anisotropy was obtained by solid-state phase transformation followed by electrochemical dealloying. The relationship between anisotropy and high-frequency magnetic loss ability was illustrated by experimental and calculational results, implying that Fe₃C flakes with a higher shape anisotropy can give a more active effect on the ferromagnetic resonance frequency. On this basis, an excellent magnetic loss ability (μ'') of 0.9 at a frequency of 11.56 GHz was achieved. The reach findings are useful for the design and preparation of high-performance microwave absorption materials.

However, there are still many issues as follow to solved before it is considered for publication. If these issues are well-addressed, this paper will be more valuable and important for the design and preparation of high-performance microwave absorption materials.

1. Is it valid to describe the "shape anisotropy" of Fe₃C only by thickness rather than width or length, since the Fe₃C in eutectoid steel presents the lamellae as illustrated in Figure 2A and 2B?
2. After etching to reveal the 2D Fe₃C flack, its thickness is not defined clearly in the text, and how to measure the thickness of Fe₃C with irregular morphology?
3. After estimating roughly, the Fe₃C thickness in Figure 2b is much larger than statistical results in Figure 2E or Figure S2.
4. By comparing Fe₃C morphology in Figure 2B, Fe₃C-700 thickness is seemingly larger than Fe₃C-500 thickness, this is invalid according $S_0=8.02 \times 10^3/\Delta T$ (line 81th).
5. The difference of "anisotropy" aroused by different isothermal quenching temperature is not presented and illustrated by XRD results Figure S5. In another words, no obvious "anisotropy effect" was achieved by the authors' preparing method, because "anisotropy effect" exerts its effect in macroscale, such as texture by rolling, single crystal by controlled directional solidification, etc.
6. Although the $M_s= 125.8$ emu/g and $H_c= 69.6$ Oe are achieved on Fe₃C-550 sample, the underlying mechanism is not adequately illustrated by current experimental or calculated results.

In summary, there is no breakthrough either in technology to prepare microwave absorption materials or in theory to design the microwave absorption materials from the current research results.

RESPONSE TO REVIEWERS' COMMENTS

Reviewer 1

Reviewer wrote:

This study presents the design of Fe₃C microflakes with high anisotropy through solid-state phase transformation, with potential application in microwave absorption. Generally speaking, this work is interesting and has some novelties in the materials design and the property simulations.

Our response: We sincerely appreciate the positive evaluation, and further we revised the whole manuscript according to the significant suggestions.

However, there are many unclear and/or ambiguous expressions in the content, likely due to a rushed submission without sufficient preparation and polishing.

Our response: We sincerely appreciate the suggestion for improving our manuscript and have polished the manuscript carefully.

The simulations are critical in understanding the effects of thickness on Fe₃C microflakes, but unfortunately, the relevant details on the simulation process are largely absent. It is therefore unable to evaluate the validation of the simulations.

Our response: We have added the details for the evaluation of validation, see the section "Theoretical calculations". (Page 15 Line 1-19)

Negative ϵ'' values can be observed in Figure S13, and most of the ϵ' values of the samples vary a lot. However, the author claims that the values of the real part (ϵ') are approximately 20 and the imaginary part (ϵ'') is 0. It seems like the declaration is not true and may mislead readers.

Our response: We have added the details for the evaluation of validation. (Page 8 Line 10-17, Ref 37-38 and the response of comment 7)

In addition, there are many obvious mistakes or typos in the manuscript. The manuscript is not well written and structured, and the corresponding mechanism is insufficiently supported by the questionable points in simulation process and experimental results.

Our response: We have added the details for the evaluation of validation.

A rejection is suggested considering the high standard of this journal.

Our response: We understand the decision of the reviewer under the consideration of

high standards and sincerely appreciate the suggestion of the reviewer for improving our manuscript. We have polished the manuscript and added the details of the simulations. Some discussion and experimental results are added to support our results. We are convinced that our revisions can meet the standard of this journal and we are writing to ask your professional advice on whether it is possible to be reconsidered in this journal.

Some specific comments:

1、 The resonance peaks in Figure 3a and b shifts a lot in the diluted (experimental) line for some of the sample, for example, Fe₃C-600 and Fe₃C-700. Please, provide a reasonable explanation. In addition, the details on the diluted plotting should be included in the figure caption,

Our response: We sincerely appreciate the suggestion. We have modified the corresponding figure caption description and added the discussion in the manuscript. In addition, the origin of the mismatch between colored and dilated lines has been described as follows. *(Page 8 Line 19-22 Page 14 Line 9-21 Ref 51-52)*

The *COLORED LINE* presented in Fig. 3 is fitted from the diluted line in the same figure. In other words, the *COLORED LINE* is calculated from the five different experimental lines through the LLG equation. We apologize for this unclear description, and we have added the description and the fitting process in the main text and method section *(Page 8 Line 19-22 Page 14 Line 9-21)*.

In addition, the details for the fitting process have been provided to make it clear.

The fitting process is made by the analysis model of OriginPro 2021, and the fitting model is the Nonlinear Curve Fit, as shown in Fig. R1.

Fig. R1. Fitting model selection of this work.

Then, the frequency is set as the Independent Variables, and permeability values (real or imaginary part) are set as the dependent Variables. The parameters a, b, and c are set as the damping parameters, initial susceptibility, and operation frequency, respectively. The fitting equation is the Landau-Lifshitz-Gilbert (LLG) equation shown as follows:

$$\mu' = 1 + \chi_0 \frac{1 + (\alpha^2 - 1) \left(\frac{f}{f_r}\right)^2}{[1 - (1 + \alpha^2) \left(\frac{f}{f_r}\right)^2]^2 + (2\alpha \frac{f}{f_r})^2} \quad (R1)$$

$$\mu'' = \chi_0 \frac{\alpha \left(\frac{f}{f_r}\right) [1 + (1 + \alpha^2) \left(\frac{f}{f_r}\right)^2]}{[1 - (1 + \alpha^2) \left(\frac{f}{f_r}\right)^2]^2 + (2\alpha \frac{f}{f_r})^2} \quad (R2)$$

After that, the fitting process will run through the equation setting based on the LLG equation. We need to select all five experimental results in the software and the fitting result would be given automatically.

Finally, we would get the fitting line which is the *COLORED LINE* in Fig. 3 (Fig. R2), and the constant including damping parameters, initial susceptibility, and operation frequency.

Fig. R2. Fitting result of the imaginary part.

Therefore, due to the line obtained by the LLG equation, their shapes would strictly follow the definition, that is, a Gaussian curve-like line for the imaginary part and a broken line-like line for the real part. Thus, the experimental line would exhibit a shift with the fitting line at the beginning and ending areas.

2、Figure 4f lacks of scale bar. What is the “colored ball” locating in top right corner of figure 4f? Please make it clear.

Our response: Figure 4f is obtained from Figures 4b and 4d, where the scale bar is consistent with Figures 4b and 4d. In the revised manuscript, we added the scale bar in Figure 4f. The “colored ball” in the top right corner denotes the in-plane magnetization, which is consistent with the white arrows in Figure 4f. We have added a detailed description in the revised manuscript. (Page 10 Line 18 Fig 4)

3、The authors should provide the detailed information (including the model, equations etc.) of the simulation in figure 5B-C, which may help to understand and evaluate the simulations.

Our response: We have added the details for the evaluation of validation, see the section of “Theoretical calculations”. (Page 15 Line 1-19)

4、The authors claim that the simulation in figure 5B-C could prove the absorption of EM for Fe₃C microflakes at high frequency could originate from the appearance of large numbers of domain walls in stripe domains. However, in my opinion, the authors should further clarify the influence of difference anisotropy by comparing the amplitudes of the

samples fabricated by different temperatures, which may support the results in Figure 5a.

Our response: The anisotropy mainly affects the position of the resonant peak, increasing the resonant frequencies. The results are consistent with the experimental data. The simulated amplitude is similar to the experimental results in Figure 5a, which has no valuable physical information. Thus, we did not present the value of the amplitude in the original submission. Anyway, we have shown the results of the experiment and simulation in Table S2. (*Table S2 in Supporting Information*)

Samples	Experimental values	Simulated values($\times 10^{-4}$)
Fe₃C-550	0.91	0.28
Fe₃C-575	0.97	NaN
Fe₃C-600	0.84	0.29
Fe₃C-625	0.86	0.13
Fe₃C-650	1.47	0.14
Fe₃C-675	1.26	NaN
Fe₃C-700	1.07	0.30

Table S2: Experimental imaginary of permeability and simulated amplitude.

5、*The damping constant of 0.1 was used for the simulations in figure 5a. Why chose 0.1? Was it from the electromagnetic parameters of Fe₃C-700?*

Our response: The damping constant is chosen by averaging the fitted damping constant in Figure 3b, as shown in Table S1. We also simulate the influence of the damping constant on dynamics. It should be noted that the change of α in the simulation only affects the intensity of amplitude in the position of resonant peaks, which does not affect the position of resonant peaks, as shown in Figure S17. The details have been added to the revised manuscript. (*Fig S17 and Table S1 in Supporting Information*)

Figure R3: The influence of damping constant on dynamics. (Fig. S17 in supporting information)

6、Figure caption of figure 5 seems not right, and the scale bars in the figures are missing.

Our response: We have revised the caption of Figure 5 and added the scale bars in the figures, see Figure 5 in the revised manuscript. (Fig 5)

7、Negative ϵ'' values can be observed in Figure S13, and most of the ϵ' values of the samples vary a lot. As a result, the following declaration is questionable: ‘For the complex permittivity (Figure S13, Supporting Information), it can be seen that the values of the real part (ϵ') are approximately 20 and the imaginary part (ϵ'') is 0. (Page 9 in the manuscript).

Our response: We agree with the reviewer that some values of complex permittivity vary from the fitting line (20 and 0). But this is not contradicted by the description in the manuscript. The details are listed below.

1. The abnormal values do not occur randomly. To demonstrate the conclusion, we have exhibited the superimposed figure of complex permittivity and complex permeability, as shown in Fig. R3. It can be recognized that the values can be found in two areas: the high-frequency turning point of the real part, and both ends of the resonance peak of the imaginary part. Meanwhile, some resonance of permeability at other regions could also

induce the inverse resonance in the permittivity (In the real part 675 and 550 samples). In addition, such results can be confirmed by the fitting lines shown in Fig. R4, in which the abnormal values can be found shifting from the fitting line at the same area.

Fig. R4. The superimposed figure of complex permittivity and complex permeability of

all five experimental results has been used in the manuscript.

2. Such a phenomenon is caused by the energy transformation between the permeability and permittivity, which can be usually found in a system with a conductive network. The corresponding reports are listed below.

A. Zhang et al. reported this transformation in the close-packed Ni nanoparticles, in which an additional dielectric loss can be found with a negative value of permeability can be noticed. This electromagnetic transform is associated with the energy transfer between both vectors of electric and magnetic fields and further gives rise to the resonance behavior when their energy states satisfy the matching with the frequency of the electromagnetic wave. (*Appl. Phys. Lett.* 97, 2010, 033107)

B. Deng et al. reported that the motion of the charges in multi-walled carbon nanotubes can radiate magnetic energy under an AC electric field and thus result in negative permeability, which is caused by the “geometrical effect”. (*Appl. Phys. Lett.* 91, 2007, 023119)

C. In the rGO system, their complex permeability could also demonstrate negative values along with the permittivity increasing. Meanwhile, it can be noticed that the permeability exhibits a symmetrical shape with the permittivity in the graphite sample. (*Appl. Phys. Lett.* 98, 2011, 072906)

D. Wu et al. found that the energy transformation between permeability and permittivity has occurred in the Ni@C-C-NbC nanocomposites, in which the magnetic behavior may be modulated by the dielectric behavior, which induces the coupling between these two parameters. (*J. Alloy. Compd.* 685, 2016, 50-57)

E. In the SiCw/wax composite prepared in the current study exhibits resonant permittivity and antiresonant permeability simultaneously. For the permeability, the locations of the maximum values and the minimum values on the μ' curve are qualitatively “reversal” to those of the ϵ' . (*Appl. Phys. Lett.* 90, 2007, 142907)

F. The $\tan \delta_E$ and $\tan \delta_M$ show a complementary variation tendency with the frequency in the high-frequency region in nickel/carbon composite microspheres. In accordance with the Maxwell formulas, the permeability, and permittivity are coupling parameters because

a magnetic field can be induced by an alternating-current electric field caused by the eddy current in EM-wave-absorbing materials and radiated out. (*ACS Appl. Mater. Interfaces*, 8, 2016, 20258-20266)

G. Gao et al. found that the sub-nanometer clusters can create a three-dimensional conductive network for conducting the free charges, which could achieve the interaction and polarization in the nanocages and thus result in the energy transformation between permeability and permittivity at 10 GHz. In this work, the addition of dielectric loss and the corresponding negative value of permeability can be found. (*Adv. Funct. Mater.* 32, 2022, 2204370)

Fig. R5. The fitting pattern of complex permittivity of all the samples (*Fig. S14* in supporting information)

In summary, it can be found that the permittivity can be reinforced with the decreased permeability due to the conductive network assembled by the samples. Therefore, the irregular shape of Fe₃C microflakes would form a conductive network in the EM testing sample, which can induce polarization around the natural resonance and thus result in the vibration of complex permittivity. Meanwhile, the corresponding discussion has been added in the manuscript, and the description of the value has been modified to “centered at 20/centered at 0” in order to prevent misunderstanding. (*Page 8 Line 10-17 Ref 37-38*)

Reviewer 2

Reviewer wrote:

In this paper, the Fe₃C with different shape anisotropy was obtained by solid-state phase transformation followed by electrochemical dealloying. The relationship between anisotropy and high-frequency magnetic loss ability was illustrated by experimental and calculational results, implying that Fe₃C flakes with a higher shape anisotropy can give a more active effect on the ferromagnetic resonance frequency. On this basis, an excellent magnetic loss ability (μ'') of 0.9 at a frequency of 11.56 GHz was achieved. The reach findings are useful for the design and preparation of high-performance microwave absorption materials.

Our response: We sincerely appreciate the very positive evaluation!

However, there are still many issues as follow to solved before it is considered for publication. If these issues are well-addressed, this paper will be more valuable and important for the design and preparation of high-performance microwave absorption materials.

Our response: We sincerely appreciate the suggestion for improving our manuscript and have polished the manuscript carefully.

1. Is it valid to describe the “shape anisotropy” of Fe₃C only by thickness rather than width or length, since the Fe₃C in eutectoid steel presents the lamellae as illustrated in Figure 2A and 2B?

Our response: Thanks for your comment. We apologize for the unclear description. The definition of the “thickness of Fe₃C” has been exhibited in Fig. S1 to eliminate the misunderstanding. Meanwhile, as Fig. 2B shows, the Fe₃C obtained from heat treatment would demonstrate an irregular lath shape. Thus, the width and length of Fe₃C would exhibit big changes. Nevertheless, it can be noticed that the thickness of Fe₃C is nanoscale but the width and length are microscale. Here, a high aspect ratio that exceeds 10⁴ can be obtained and thus constructed a high shape anisotropy to enhance the microwave performance. The details have been added to the manuscript. (Page 3 Line

22-24, Page 6 Line 7-9, Fig. S1).

Fig. R6. Schematic diagram of the definition in the main text. (Fig. S1 in supporting information)

2. After etching to reveal the 2D Fe₃C flack, its thickness is not defined clearly in the text, and how to measure the thickness of Fe₃C with irregular morphology?

Our response: We apologize for the unclear description. The definition of the “thickness of Fe₃C” has been exhibited in Fig. S1 to eliminate the misunderstanding. The details have been added to the manuscript. (Page 13 Line 17-19, Fig. S1).

3. After estimating roughly, the Fe₃C thickness in Figure 2b is much larger than statistical results in Figure 2E or Figure S2.

Our response: Thanks for your comment. The TEM (Fig. 2B) analysis direction has been demonstrated in Fig. S1. The thickness can not be detected in TEM images and we apologize for the unclear description. The statistic of the micro-flake thickness is calculated via the SEM images using the open-access *ImageJ* software. (Page 13 Line 17-19, Fig. S1)

4. By comparing Fe₃C morphology in Figure 2B, Fe₃C-700 thickness is seemingly larger than Fe₃C-500 thickness, this is invalid according $S_0=8.02 \times 103/\Delta T$ (line 81th).

Our response: We apologize for the unclear description. The definition of ΔT means the difference between the heat treatment temperature (800 °C) and the isothermal quenching temperature. Therefore, the S_0 of Fe₃C-700 is larger than Fe₃C-550. The detail has been added to the manuscript. (Page 3 Line 26-27).

5. The difference of “anisotropy” aroused by different isothermal quenching temperature is not presented and illustrated by XRD results Figure S5. In another words, no obvious “anisotropy effect” was achieved by the authors’ preparing method,

because “anisotropy effect” exerts its effect in macroscale, such as texture by rolling, single crystal by controlled directional solidification, etc.

Our response: Thanks for your advice. The anisotropy in this work refers to shape anisotropy, which is manipulated by the flake geometry to break the “Snoek limit” and thus improve the natural resonance performance of magnetic loss materials. (Page 2 Line 16-18). In this work, the shape anisotropy of Fe₃C has been changed by the isothermal quenching process, because the thickness of Fe₃C can be reduced along with the quenching temperature decreased. (Page 3 Line 26-27) To demonstrate the effect of shape anisotropy, we first count the thickness of Fe₃C to prove the changing of thickness. (Fig. 2E) Then, we analyzed the electromagnetic response performance, which indicated that the natural resonance frequency had increased to 11.56 GHz. Ascribing to the only changing conditions is the thickness of Fe₃C, it can be implied that the reduced thickness could enhance the shape anisotropy and thus improve the electromagnetic performance. (Page 9 Line 5-9) Meanwhile, we used theoretical simulation to certify the relationship. As Fig. 5A shows, the ferromagnetic resonance spectra of the Fe₃C microflakes under different temperatures after obtaining the static magnetic structures with different values of K_i , which are induced by the shape anisotropy of Fe₃C microflakes with different thicknesses. (Page 11 Line 10-12) Here, the effect of shape anisotropy for improving the ferromagnetic loss performance has been evidenced.

6. Although the $M_s = 125.8 \text{ emu/g}$ and $H_c = 69.6 \text{ Oe}$ are achieved on Fe₃C-550 sample, the underlying mechanism is not adequately illustrated by current experimental or calculated results.

Our response: According to the “Snoek limit” ($(\mu_i - 1)f_r = 4\gamma M_s/3$), it can be noticed that the initial permeability μ_i and natural resonance frequency f_r could not be simultaneously enhanced because the saturation magnetization could not be changed at will. In this work, it can be noticed that the value of M_s stays in close proximity, which means that the increased natural resonance performance can not be contributed by the saturation magnetizations. Meanwhile, the H_c has been increased with the isothermal quenching decreased to 550 °C. The coercivity is proportional to the anisotropy constant, can be inferred that the increase in coercivity is the consequence of the

increased anisotropies of the 2D Fe₃C microflakes (*Acta Mater.* 231, 2022, 117854).
(Page 7 Line 9-11, Ref. 35)

In summary, there is no breakthrough either in technology to prepare microwave absorption materials or in theory to design the microwave absorption materials from the current research results.

Our response: Thanks for your advice. The microwave absorption property is mainly dominated by the natural resonance for magnetic loss materials at GHz. However, the Snoek limit ($(\mu_i - 1)f_r = 4\gamma M_s / 3$) becomes the fatal issue that hinders the steps of soft magnetic materials toward high-frequency applications in which initial permeability μ_i and natural resonance frequency f_r could not be simultaneously enhanced because the saturation magnetization could not be changed at will. (Page 2 Line 16-18) By introducing the magnetic anisotropy in magnetic materials (including shape anisotropy, plane anisotropy), the “Snoek limit” equation can be rewritten to $(\mu_i - 1)f_r = (\gamma M_s / 2)(|H_\theta|/|H_\phi|)^{1/2}$. Thus, the resonance frequency and permeability can be optimized further. Some attempts have been reported to increase the performance through shape anisotropy, but the traditional method could not construct an enough aspect ratio to increase the resonance frequency above 10 GHz.

In this work, by using the electrochemical dealloying method, we can modify the shape of Fe₃C microflakes and thus increase their shape anisotropy. To demonstrate the effect of shape anisotropy, we first count the thickness of Fe₃C to prove the changing of thickness. (Page 3 Line 26-27) It can be noticed that the thickness of Fe₃C is nanoscale but the width and length are microscale. Here, a high aspect ratio that exceeds 10⁴ can be obtained and thus constructed a high shape anisotropy to enhance the microwave performance. Then, we analyzed the electromagnetic response performance, which indicated that the natural resonance frequency had increased to 11.56 GHz. Ascribing to the only changing conditions is the thickness of Fe₃C, it can be implied that the reduced thickness could enhance the shape anisotropy and thus improve the electromagnetic performance. (Page 9 Line 5-9) Meanwhile, we used theoretical simulation to certify the relationship. As Fig. 5A shows, the ferromagnetic resonance spectra of the Fe₃C microflakes under different temperatures after obtaining the static magnetic structures

with different values of K_i , which are induced by the shape anisotropy of Fe_3C microflakes with different thicknesses. *(Page 11 Line 10-12)* The present study provides an intrinsic insight into the role of anisotropy in high-frequency magnetic loss ability. Furthermore, this method can be easily expanded to macro preparation, which promises an engineering application potential. *(Page 12 Line 21-24)*

REVIEWER COMMENTS

Reviewer #1 (Remarks to the Author):

The authors have responded quite diligently to our questions, but I believe there are still a few issues that have not been addressed. Furthermore, the primary concept of this work, which involves controlling anisotropy through shape modification to affect the natural or exchange resonance frequencies and microwave absorption of materials, has been explored in previous studies and yields largely foreseeable results. Consequently, we still do not believe that the methods employed and the results obtained in this work represent significant breakthroughs.

1. The authors have included the simulation details in the revised manuscript, as previously requested. However, it has come to our attention that the fitting process using the LLG equation is not particularly novel.
2. The response of comment 5 is not solid and contains some evident errors. For example, the authors claim that the change of α in the simulation solely impacts the intensity of amplitude at the resonant peak positions, without affecting the actual positions of the resonant peaks. However, as illustrated in Figure S17, α not only determines the number of resonance peaks but also determines the presence of peaks. Two resonance peaks are observed when $\alpha=0.01$ and 0.02 , one peak is observed when $\alpha=0.1$ and 0.2 , and no peak is observed when $\alpha=0.5$. Therefore, the validity of the simulation is questionable, or at the very least the presentation is not clear.
3. The response in comment 7 is insufficient. The fitting line lacks solidity due to arbitrary values used for the intensities of ϵ' and ϵ'' . This obscures the distinctions in the dielectric properties of the materials and may mislead readers' understanding of the materials.

Reviewer #2 (Remarks to the Author):

In the current paper, the authors have revised clearly all the questionable matters. However, to highlight the innovation and readability of manuscript, a few suggestions are presented as followed:

1. The positions of Figure 2c and 2d were replaced with Figure S1 and S5. And the Figure 2c and 2d can be considered as supporting information, because the Fe₃C information can be readily recognized anywhere.
2. Please supply the images marked 1, 2, 3 and 4 in Figure S4 with higher magnification to reveal the morphology of thickness of Fe₃C in nanoscale.

RESPONSE TO REVIEWERS' COMMENTS

Reviewer 1

Reviewer wrote:

The authors have responded quite diligently to our questions, but I believe there are still a few issues that have not been addressed. Furthermore, the primary concept of this work, which involves controlling anisotropy through shape modification to affect the natural or exchange resonance frequencies and microwave absorption of materials, has been explored in previous studies and yields largely foreseeable results. Consequently, we still do not believe that the methods employed and the results obtained in this work represent significant breakthroughs.

Our response: We sincerely appreciate the positive evaluation, and further we revised the whole manuscript according to the significant suggestions.

It is true that the influence of shape anisotropy on microwave absorption has been explored in previous studies. However, in this work, we present the design of Fe₃C microflakes through a new method (solid-state phase transformation). By using this method, we achieved the preparation of functional materials from traditional structural materials (steel), which could stimulate the research interests in obtaining functional materials from other traditional structural materials. We believe that this work also provides some preliminary research basis for the realization of structural-functional integrated materials. We have added a corresponding description in the revised manuscript to emphasize the novelty. *(Page 1, Lines 28-29; Page 13, Lines 1-3)*

1. The authors have included the simulation details in the revised manuscript, as previously requested. However, it has come to our attention that the fitting process using the LLG equation is not particularly novel.

Our response: Thanks for your comment. The LLG equation is one of the classical mathematical equations in magnetism, which is significant from the point of view of fundamental physics and practical applications for devices. Herein, the LLG equation is used as a powerful tool to assertively explain the physical mechanism of excellent microwave absorption performance. The novelty of this work is the design of Fe₃C microflakes through solid-state phase transformation. *(Page 1, Lines 28-29; Page 14,*

Lines 19-20)

For the measured complex permeability experimentally, we have fitted it by the LLG equation. On the one hand, the damping constant can be obtained, which is used in micromagnetic simulations to explore the influence of anisotropy on ferromagnetic resonances. On the other hand, the fitting process can be used to rule out accidental errors that might occur in the experiment and is statistically significant.

Then, micromagnetic simulations are performed to explore the influence of anisotropy on the ferromagnetic resonances, which is based on the solving of the LLG equation numerically. It is inferred that the increased resonant frequencies can be ascribed to the increase in effective in-plane anisotropy, which can be manipulated by adjusting the isothermal quenching temperatures.

2. The response of comment 5 is not solid and contains some evident errors. For example, the authors claim that the change of α in the simulation solely impacts the intensity of amplitude at the resonant peak positions, without affecting the actual positions of the resonant peaks. However, as illustrated in Figure S17, α not only determines the number of resonance peaks but also determines the presence of peaks. Two resonance peaks are observed when $\alpha=0.01$ and 0.02 , one peak is observed when $\alpha=0.1$ and 0.2 , and no peak is observed when $\alpha=0.5$. Therefore, the validity of the simulation is questionable, or at the very least the presentation is not clear.

Our response: Thanks for reminding us of the unclear description. Because the damping constant can affect the amplitude of resonant peaks, some other spin wave modes can be found under smaller damping constants and the main spin wave modes can be hidden under larger damping constants. For example, for research about the propagation of spin waves, the damping constant is usually set at a smaller value to explore comprehensive physical phenomena. Meanwhile, a higher damping constant is set near the edge to annihilate the undesired scattered waves [Nat. Commun. 9(2018) 4853].

In this work, we obtain the damping constant from experimental results and the value is 0.1, where one main peak is found. Then, we study the influence of anisotropy on ferromagnetic resonances by fixing the damping constant at 0.1. We have revised the description in the revised manuscript. (*Page 15, Lines 9-12, Ref. 53, and the caption of*

Figure S15 in Supporting Information)

3. The response in comment 7 is insufficient. The fitting line lacks solidity due to arbitrary values used for the intensities of ϵ' and ϵ'' . This obscures the distinctions in the dielectric properties of the materials and may mislead readers' understanding of the materials.

Our response: Thanks for your comment. To eliminate the misunderstanding of the dielectric ability of our materials, the complex permittivity with fitting line has been removed from the supporting information. Meanwhile, the superimposed figure of complex permittivity and complex permeability (demonstrated as *Fig. R4* in the last revision) has been added to the supporting information. (*Figure S12 in Supporting Information*) In addition, the discussion of dielectric ability has been added to the revised manuscript to further explain the resonance of complex permittivity. We believe the corresponding revision could explain the phenomenon of the dielectric ability of our materials which could also let readers understand the transformation between the permittivity and permeability. (*Page 8 Line 7-17 & 21-23 and Figure S12 in Supporting Information*)

Reviewer 2

Reviewer wrote:

In the current paper, the authors have revised clearly all the questionable matters. However, to highlight the innovation and readability of manuscript, a few suggestions are presented as followed:

Our response: We sincerely appreciate the very positive evaluation!

1. The positions of *Figure 2c* and *2d* were replaced with *Figure S1* and *S5*. And the *Figure 2c* and *2d* can be considered as supporting information, because the *Fe₃C* information can be readily recognized anywhere.

Our response: Thanks for your comment. The corresponding figures have been modified in the revised manuscript. (*Figure 2 and Figure S6 in Supporting Information*)

2. Please supply the images marked 1, 2, 3 and 4 in *Figure S4* with higher magnification

to reveal the morphology of thickness of Fe₃C in nanoscale.

Our response: Thanks for your comment. The corresponding figures have been added to the revised manuscript. (*Figure S3 in Supporting Information*)

REVIEWERS' COMMENTS

Reviewer #1 (Remarks to the Author):

Thanks to the author for patiently addressing our questions. However, despite the author's diligent efforts in answering my queries, we still have reservations about the innovative aspect of the paper. For instance, we question whether the "design of Fe₃C microflakes through solid-state phase transformation" can serve as a primary innovative point to support the paper's publication in NC. Considering my previous questions and the author's responses, my current conclusion is that the paper can be accepted for publication, but I still maintain my own opinion.

RESPONSE TO REVIEWERS' COMMENTS

Reviewer 1

Reviewer wrote:

Thanks to the author for patiently addressing our questions. However, despite the author's diligent efforts in answering my queries, we still have reservations about the innovative aspect of the paper. For instance, we question whether the "design of Fe₃C microflakes through solid-state phase transformation" can serve as a primary innovative point to support the paper's publication in NC. Considering my previous questions and the author's responses, my current conclusion is that the paper can be accepted for publication, but I still maintain my own opinion.

Our response: We sincerely appreciate your evaluation, and your comments have helped us tremendously to refine our work. We believe this work provides some preliminary research basis for the realization of structural-functional integrated materials and which may provide a pathway to design functional materials from traditional structural materials.